# Targeting HSF1 as a Therapeutic Strategy for Multiple Mechanisms of EGFR Inhibitor Resistance in EGFR Mutant Non-Small-Cell Lung Cancer

**DOI:** 10.3390/cancers13122987

**Published:** 2021-06-15

**Authors:** Sangah Lee, Jiyae Jung, Yu-Jin Lee, Seon-Kyu Kim, Jung-Ae Kim, Bo-Kyung Kim, Kyung Chan Park, Byoung-Mog Kwon, Dong Cho Han

**Affiliations:** 1Personalized Genomic Medicine Research Center, Korea Research Institute of Bioscience and Biotechnology, 111 Gwahangno, Yuseong-gu, Daejeon 34141, Korea; sal22@kribb.re.kr (S.L.); jjy0510@kribb.re.kr (J.J.); seonkyu@kribb.re.kr (S.-K.K.); jungaekim@kribb.re.kr (J.-A.K.); kimbk@kribb.re.kr (B.-K.K.); kpark@kribb.re.kr (K.C.P.); 2KRIBB School of Bioscience, University of Science and Technology in Korea, 111 Gwahangno, Yuseong-gu, Daejeon 34141, Korea; 3Genome Editing Research Center, Korea Research Institute of Bioscience and Biotechnology, 111 Gwahangno, Yuseong-gu, Daejeon 34141, Korea; yujini@kribb.re.kr

**Keywords:** EGFR, resistance, HSF1, inhibitors, NSCLC

## Abstract

**Simple Summary:**

We attempted to identify target proteins and compounds that can be used to overcome EGFR-TKI resistance in NSCLC. To accomplish this, we generated EGFR inhibitor erlotinib-resistant HCC827-ErlR cells and obtained a list of differentially expressed genes. Then, we performed connectivity map analysis and identified heat shock factor 1 (HSF1) as a potential target protein to overcome erlotinib resistance. Using specific HSF1 shRNAs and KRIBB11 (N^2^-(1H-Indazol-5-yl)-N^6^-methyl-3-nitropyridine-2,6-diamine), we proved the effectiveness of HSF1 inhibition for overcoming erlotinib resistance in vitro. In addition, we proved the efficacy of emetine in inhibiting HSF1 activity and the tumor growth of erlotinib-resistant PC9-ErlR cells in a mouse model.

**Abstract:**

Although EGFR-TKI treatment of NSCLC (non-small-cell lung cancer) patients often achieves profound initial responses, the efficacy is transient due to acquired resistance. Multiple receptor tyrosine kinase (RTK) pathways contribute to the resistance of NSCLC to first- and third-generation EGFR-TKIs, such as erlotinib and osimertinib. To identify potential targets for overcoming EGFR-TKI resistance, we performed a gene expression signature-based strategy using connectivity map (CMap) analysis. We generated erlotinib-resistant HCC827-ErlR cells, which showed resistance to erlotinib, gefitinib, osimertinib, and doxorubicin. A list of differentially expressed genes (DEGs) in HCC827-ErlR cells was generated and queried using CMap analysis. Analysis of the top 4 compounds from the CMap list suggested HSF1 as a potential target to overcome EGFR-TKI resistance. HSF1 inhibition by using HSF1 shRNAs or KRIBB11 decreased the expression of HSF1 downstream proteins, such as HSP70 and HSP27, and also decreased the expression of HSP90/HSP70/BAG3 client proteins, such as BCL2, MCL1, EGFR, MET, and AXL, causing apoptosis of EGFR-TKI-resistant cancer cells. Finally, we demonstrated the efficacy of the HSF1 inhibitor on PC9-ErlR cells expressing mutant EGFR (T790M) in vivo. Collectively, these findings support a targetable HSF1-(HSP90/HSP70/BAG3)-(BCL2/MCL1/EGFR/MET/AXL) pathway to overcome multiple mechanisms of EGFR-TKI resistance.

## 1. Introduction

Lung cancer is categorized as either small-cell lung cancer (SCLC, comprising 20% of lung cancer cases) or non-small-cell lung cancer (NSCLC, comprising 80% of lung cancer cases) and has the highest cancer incidence and mortality worldwide [1]. Treatment with EGFR-TKIs (tyrosine kinase inhibitors) is the standard of care for NSCLC patients with activating EGFR mutations. Approximately 15% of Caucasian patients and 50% of Asian patients have activating EGFR mutations [2], and treatment of these patients with first-generation EGFR-TKIs (gefitinib or erlotinib) significantly increases overall survival [3,4,5,6].

Unfortunately, most patients treated with the first-line treatment of EGFR-TKIs develop resistance within 13 months [7]. The molecular mechanisms of acquired resistance to EGFR-TKIs include drug target alterations, such as the EGFR (T790M) mutation, or activation of redundant pathways, such as MET amplification [8,9]. The EGFR (T790M) mutation is the most common cause of acquired resistance to EGFR-TKIs and is found in approximately 60% of NSCLC patients treated with first-generation EGFR-TKIs. Therefore, third-generation EGFR-TKIs (osimertinib, olmutinib, and rociletinib) were developed to treat NSCLC patients with the EGFR (T790M) mutation [10,11,12]. Unfortunately, pharmaceutical development of rociletinib and olmutinib has ceased [13]. In contrast, osimertinib has been successfully developed and has demonstrated a significant progression-free survival benefit in NSCLC patients with the EGFR (T790M) mutation [14]. Progression free survival on osimertinib first line is 18.9 months versus 10.2 month for other first-generation EGFR-TKIs [15].

Osimertinib has been approved as an EGFR-TKI for the EGFR (T790M) mutation, but resistance to osimertinib remains problematic [16]. Analysis of DNA sequences from EGFR (T790M) patients with acquired resistance to osimertinib revealed the presence of a novel EGFR (C797S) mutation. The C797S mutation eliminates the drug binding site because osimertinib covalently binds to the cysteine at residue 797. One-quarter of EGFR (T790M)-positive tumors that acquire resistance to osimertinib lose the T790M mutation [16]. The resistance mechanisms of T790M-negative tumors include the amplification of MET or HER2, the epithelial-to-mesenchymal transition (EMT), phenotype change to small-cell lung cancer (SCLC), and point mutations in B-RAF, KRAS, TP53, and PIK3CA (for review, [17]).

Analysis of circulating tumor DNA from NSCLC patients revealed that 46% of patients treated with EGFR inhibitors have multiple drug resistance mechanisms [18]. More than half of erlotinib-resistant cells acquire the EGFR (T790M) mutation. However, MET amplification coexists with or without the T790M mutation in EGFR mutant lung tumors [19]. The co-occurrence of the T790M mutation and other mechanisms of resistance to the primary EGFR-TKI is much greater than previously reported, and such intrapatient heterogeneity may affect the clinical response to subsequent EGFR-TKIs. Thus, single oncogene-targeted therapy may have incomplete responses due to resistance acquired simultaneously by multiple mechanisms.

HSF1 is a master transcription factor for the expression of heat shock proteins (HSPs), such as HSP70, HSP27, and BAG3. Under nonstress conditions, monomeric HSF1 binds to the HSP90/HSP70 chaperone complex [20]. When cells are exposed to proteotoxic stress, HSP90 and HSP70 are displaced from the chaperone complex, releasing HSF1 [20,21,22]. Free HSF1 then translocates into the nucleus and trimerizes and binds to heat shock elements in the promoter regions of target genes [23]. Tumorigenesis in several animal models was shown to be HSF1 dependent, and HSF1 knockout significantly decreased tumor formation and progression [24,25]. HSP90 chaperone complexes regulate the stability and function of diverse client proteins, including EGFR [26,27], MET [28], and AXL [29,30]. Importantly, HSP70 also functions as a cochaperone for HSP90 [31,32]. The HSC70/HSP90-organizing protein HOP binds to both HSP70 and HSP90 and transfers clients from HSP70 to HSP90, thus functioning as an adaptor for the two chaperones [33,34]. Dual targeting of HSC70 and HSP72 inhibits HSP90 function and induces tumor-specific apoptosis [35]. Similarly, the depletion of HSP70 activates a tumor-specific death mechanism that is independent of caspases and bypasses BCL2 [36], suggesting that HSP70 and HSP90 are potential targets to induce apoptosis of cancer cells. In addition to the classical role of HSF1 in protein homeostasis, HSF1 transcriptionally regulates a distinct pathway that supports malignant transformation and cancer cell survival and proliferation [37].

The connectivity map (CMap) is a big data collection of transcriptome alterations in response to various small molecules that has been used experimentally and clinically in human cancer cell lines, and it provides pattern-matching web-based software to mine these data, identifying biologically active compounds with similar or opposite activity [38,39]. As most CMap compounds are FDA-approved drugs, these analyses have become a valuable tool for understanding drug mechanisms of action and for drug repurposing. In this study, we sought to identify the therapeutic vulnerability of erlotinib-resistant HCC827-ErlR cells and identified HSF1 as a potential target to overcome EGFR-TKI resistance in NSCLC tumors.

## 2. Materials and Methods

### 2.1. Reagents

Unless otherwise specified, all chemicals in the study, including DMSO (dimethyl sulfoxide) and emetine, were purchased from Sigma (St Louis, MO, USA). Erlotinib, gefitinib, doxorubicin, and cephaeline were purchased from Cayman Chemical (Ann Arbor, MI, USA). KRIBB11 and osimeritnib were purchased from Selleckchem (Houston, TX, USA).

Antibodies against HSF1 (#12972), HSP27 (#2402), HSP90α (#8165), PARP (#8165), phospho-ERK (#9101), ERK (#4695), phospho-MEK (#9121), MEK (#9122), BCL2 (#15071), MCL1 (#4572), caspase-3 (#9662), cleaved caspase-3 (#9664), PARP (#9542), EGFR (#4267), phospho-EGFR (Y1068) (#3777), MET (#8198), HER2 (#4190), and vimentin (#5741) were purchased from Cell Signaling Technology (Danvers, MA, USA), and actin antibody (sc-47778) was purchased from Santa Cruz Biotechnology (Dallas, TX, USA). Antibodies against phospho-HSF1 (Ser326) (#ab76076), BAG3 (ab47124) and E-cadherin (ab15148) were purchased from Abcam (Cambridge, MA, USA). N-cadherin antibody (640920) was purchased from BD Bioscience (San Jose, CA, USA). Horseradish peroxidase-conjugated goat anti-rabbit, mouse anti-rat, and goat anti-mouse IgG antibodies were obtained from Jackson ImmunoResearch (West Grove, PA, USA).

### 2.2. Cell Culture

HCC827 (CRL-2868), NCI-H820 (HTB-181), and HCT116 (CCL-247) cell lines were obtained from ATCC (Manassas, VA, USA). PC9 (CVCL_B260) cells were obtained from Prof. Chaeuk Chung (Department of Internal Medicine, Chungnam National University Hospital, Daejeon, Korea). HCC827-ErlR cells were developed from HCC827 cells by exposing cells to 1 μM erlotinib for 3 months. PC9-ErlR cells were developed from PC9 cells by a stepwise increase in erlotinib concentrations from 5 nM to 1 μM for 3 months. All cell lines were maintained in RPMI 1640 supplemented with 10% heat-inactivated fetal bovine serum (FBS) (Invitrogen, Waltham, MA, USA). Cell cultures were maintained at 37 °C in 5% CO_2_ in an incubator.

### 2.3. mRNA Expression Profiling and Connectivity Map Analysis

RNA was isolated from HCC827 and HCC827-ErlR cells using the RNeasy Mini Kit (Qiagen, Valencia, CA, USA). The quality and quantity of the samples for sequencing were evaluated using the Agilent 2100 Bioanalyzer (Agilent Technologies, CA, USA) and the μDropTM (Thermo Scientific, Waltham, MA, USA). An RNA library was generated using the TruSeq RNA Sample Prep Kit v2 (Illumina, San Diego, CA, USA) according to the manufacturer’s instructions. The mRNA was purified from extracted total RNA using oligo-dT-attached magnetic beads and fragmented into small pieces. The cleaved RNA fragments were copied into first strand cDNA using reverse transcriptase and random primers. Second strand cDNA synthesis was then performed using DNA polymerase I and RNase H. The cDNA fragments then underwent an end repair process, which included the addition of a single ‘A’ base and ligation of the adapters. The product was purified and enriched by PCR to generate the final cDNA library. Sequencing of the cDNA libraries was performed by the Illumina HiSeq 4000 (San Diego, CA, USA) at Macrogen (Seoul, Korea). The RNA-seq dataset was deposited in the NCBI’s Sequence Read Archive (accession number: PRJNA730670).

For the CMap build 02 analyses, the DEGs (Appendix A) from HCC827 and HCC827-ErlR cancer cell lines were determined using the FPKM RNA_seq raw data. The cutoff criteria for DEGs were set to log2(FC) > 2.0 and log2(FC) < −2.7. DEG files were prepared using an in-house script (https://github.com/a00101/convert_GRP, 4 December 2017). Then, these probe sets were queried according to the instructions. These DEGs (Grp files) were directly inputted into the Broad Institute website (https://portals.broadinstitute.org/, 4 December 2017). The ranking was determined based on the similarity of the perturbagen signature to the query signature.

### 2.4. Cell Proliferation Assay

Cells were seeded into 96-well plates at a density of 3 × 10^3^ cells per well in RPMI 1640 medium with 10% FBS. After 18 h, the medium was replenished with fresh complete medium containing chemicals or 0.1% DMSO. After incubation for 48 h or 72 h, cells were counted using a microscope.

### 2.5. Knockdown of HSF1 Using shRNA

Scrambled shRNA (Addgene #10878) and HSF1 shRNAs (Clone NM_005526. TRCN0000007480 and NM_005526. TRCN0000007482) in the pLKO.1 vector backbone, psPAX2, and pMD2. G vectors (Addgene, Cambridge, MA, USA) were cotransfected into HEK293T cells. Virus supernatants were collected 48 h after transfection, and the media was filtered through a 0.45 μm filter and transferred to HCC827-ErlR cells at a density of 3 × 10^5^ cells per well in 60 mm dishes.

### 2.6. Western Blotting

Lysates were prepared using RIPA buffer as described previously [40]. Total protein concentrations were measured with the Bradford protein assay. Next, 20 μg of protein was subjected to SDS-PAGE and transferred to PVDF membranes (Millipore, Burlington, MA, USA). Proteins were detected with the indicated primary antibodies. The secondary antibodies used were horseradish peroxidase-conjugated goat anti-rabbit, mouse anti-rat, and goat anti-mouse IgG (Jackson Immunoresearch Laboratories, Inc., West Grove, PA, USA). The blots were developed with an enhanced chemiluminescence detection reagent (Millipore, Burlington, MA, USA), and the signal was detected using the LAS 4000 mini system (GE Healthcare Life Sciences, Pittsburgh, PA, USA). Densitometric analysis of the bands was performed using Multigauge software (Fuji Photo Film Co, Ltd., Tokyo, Japan), and the results were normalized to the corresponding actin level.

### 2.7. Transwell Migration and Invasion Assay

Migration assays were performed as previously described [41]. Cell migration assays were performed using 8.0 μm pore size Transwell inserts (BD Biosciences, San Jose, CA, USA). A total of 1.5 × 10^4^ cells in 0.2 mL of serum-free medium were added to the upper chamber, and 0.5 mL of medium with 10% FBS was placed in the lower chamber. After 18 h, the migrated cells attached to the lower surface were stained with crystal violet (500 μL of 5 mg/mL crystal violet dissolved in 20% methanol) (Sigma-Aldrich, St Louis, MO, USA) and incubated for 10 min. The membrane was washed 2 times with PBS, and the cells passing through the filter were counted under a microscope (Nikon Eclips TE300; Nikon, Tokyo, Japan). For the invasion assay, Matrigel basement membrane matrix (Corning, Corning, NY, USA) was diluted 1/5 with serum-free medium using a cooled pipette and was coated inside the inserts in a volume of 200 μL. After incubation for 1 h on a clean bench, the unbound material was aspirated. The inside of the inserts was gently rinsed with serum-free medium and used for assays.

### 2.8. Luciferase Reporter Construct and Dual-Luciferase Reporter Assay

HSF1 reporter assays were carried out as previously described [42]. The p(HSE)4-TA-Luc plasmid has four copies of palindromic HSE (5′-GAT CTA GAA CGT TCT AGA ACG TTC TAG AAC GTT CTA-3′) in front of the pTA-Luc promoter vector (Clontech, CA, USA). The activity of the reporter was measured using the Dual-Luciferase^®^ Reporter Assay System (Promega, Madison, WI, USA). HCT-116 cells were cotransfected with 9 μg of p(HSE)4-TA-Luc vector and 1 μg of pRL-TK containing the Renilla luciferase gene as an internal control vector. Five hours after transfection, cells were trypsinized and seeded onto sterilized, black-bottom, 96-well plates at a density of 2 × 10^4^ cells per well. After incubation for 24 h, cells were pretreated with chemicals for 30 min, exposed to heat shock at 42 °C for 30 min, and then incubated further at 37 °C for 5 h. Firefly and Renilla luciferase activities were measured using a Dual-Luciferase^®^ Reporter Assay kit (Promega, Madison, WI, USA) with a GloMaxTM 96 Microplate Luminometer (Promega, Madison, WI, USA).

### 2.9. FACS Analysis for Apoptosis

HCC827 or HCC827-ErlR cells were treated with KRIBB11 at 5 μM and 10 μM for 48 h. Cells were carefully resuspended and treated with annexin-V (5 μL) and propidium iodide (PI) (5 μL) (BD Biosciences, San Jose, CA, USA) and then incubated at 37 °C for 15 min in the dark. The stained cells were analyzed using a FACSCalibur flow cytometer (BD Bioscience).

### 2.10. Sequencing of the EGFR Exon 20 (T790) Region

PC9-ElrR cells were plated in 96-well plates at a density of 0.5 cells per well. After a single colony was formed in each well, we selected and grew 10 subclones from PC9-ErlR cells. For sequencing, each subclone was plated in a 60 mm dish at a density of 4 × 10^5^ cells per well. After 24 h, the total RNA was isolated using the RNeasy kit (Qiagen, Hilden, Germany).

For each subclone, 1 microgram of isolated RNA was reverse transcribed with the RevertAid First Strand cDNA Synthesis Kit (Toyobo, Osaka, Japan) according to the manufacturer’s instructions. Thereafter, the EGFR exon 20 region was amplified by PCR using EGFR-specific primers, and the PCR product was sequenced by Solgent Co. (Daejeon, Korea). The following primers were used for PCR amplification: EGFR exon 20 forward primer, 5′-CCCAACCAAGCTCTCTTGAG-3′, and EGFR exon 20 reverse primer, 5′-ATGACAAGGTAGCGCTGGGG-3′.

For HCC827-ErlR cells, the total RNA was isolated, and 1 microgram of mRNA was reverse transcribed with the RevertAid First Strand cDNA Synthesis Kit (Toyobo, Osaka, Japan). Then, the exon 20 region was PCR amplified using EGFR-specific primers, and the PCR products were ligated into the T-Vector pMD20 (#3270, Takara, Kusatsu, Japan). After transforming *E. coli* DH5α cells with plasmid ligation products, 10 single colonies were selected, and plasmids were isolated and sequenced.

### 2.11. Nude Mouse Xenograft Assay

All animal work was performed in accordance with the guidelines and under the approval of the Institutional Review Committee for Animal Care and Use at the Korea Research Institute of Bioscience and Biotechnology (Approval code number: KRIBB-AEC-19024; date of approval, 24 January 2019). All animals were housed in a pathogen-free animal facility at the Korea Research Institute of Bioscience and Biotechnology (KRIBB). Seven-week-old female inbred specific-pathogen-free (SPF) Balb/c nude mice were obtained from Nara Biotech (Seoul, Korea), housed under sterile conditions with 12 h light/dark cycles, and fed food and water ad libitum. To evaluate the antitumor activity of emetine in vivo, PC9-ErlR cells (0.3 mL of 9 × 10^6^ cells/mouse) were implanted subcutaneously into the right flank of the mice. Emetine (0.1 mg/mL or 1 mg/mL) was dissolved in filter-sterilized PBS. When tumor volumes reached 55 mm^3^ (day 1), control PBS solution (200 μL) or emetine (1 mg/kg) was injected intraperitoneally for 5 days. Beginning on day 7, emetine (10 mg/kg) was injected 5 days per week for 18 days. On day 25, the mice were sacrificed, and the tumors were removed and weighed (mg). Tumor volume was estimated by using the formula: length (mm) × width (mm) × height (mm)/2. Tumor measurements were performed every 3 days with a caliper. To determine the toxicity of the compound, the weight of tumor-bearing animals was recorded.

### 2.12. Statistical Analysis

All experiments were performed at least twice, and multiple samples represented biological (not technological) replicates. All animal experiments were performed using randomly assigned mice without investigator blinding. Data are expressed as the mean ± standard deviation (S.D.), and significance was analyzed using a Student’s *t*-test. A *p*-value less than 0.05 was considered to be indicative of statistical significance. *, *p* < 0.05 compared with the control and **, *p* < 0.01 compared with the control. All tested concentrations were compared to the vehicle controls to obtain p-values using a Student’s t-test. The half maximal inhibitory concentration (IC_50_ for growth, GI_50_) values were determined from plots made using the Logistic’s curve model, and 4 parameters were determined using SigmaPlot software (Systat Software, Inc., San Jose, CA, USA).

## 3. Results

### 3.1. Generation and Characterization of HCC827-ErlR Cancer Cells

HCC827 and PC9 cells were derived from Caucasian female and Japanese male tumors, respectively, with a mutation in the EGFR tyrosine kinase domain (deletion of exon 19; E746-A750), which confers sensitivity to EGFR-TKIs. We generated two erlotinib-resistant cell lines, HCC827-ErlR and PC9-ErlR, by treating HCC827 and PC9 cells with erlotinib as described in the Materials and Methods. To analyze the effects of erlotinib on the proliferation of HCC827-ErlR and PC9-ErlR cells, we treated these cells with erlotinib at different concentrations (0–100 μM) for 72 h. As shown in Figure 1A,B, the GI_50_ values of erlotinib for HCC827-ErlR and PC9-ErlR cells were 1.97 μM and 3.98 μM, respectively, which were 492 and 1990 times higher than those of the HCC827 and PC9 parental cells. GI_50_ is the inhibitor concentration at which 50% inhibition of cell growth is seen. HCC827-ErlR and PC9-ErlR cells also showed resistance to gefitinib.

Interestingly, PC9-ErlR cells were resistant to first-generation EGFR-TKIs but sensitive to the third-generation EGFR-TKI osimertinib (Table 1), suggesting that PC9-ErlR cells acquired erlotinib resistance through an EGFR (T790M) mutation. Sequencing analysis of the EGFR (T790) region showed that PC9-ErlR cells had an EGFR (T790M) mutation (Appendix A). In contrast, HCC827-ErlR cells showed resistance to erlotinib and osimertinib (Table 1), suggesting no EGFR (T790M) mutation in HCC827-ErlR cells. This interpretation was consistent with DNA sequencing results in the EGFR (T790) region (Appendix A). In addition, the GI_50_ value of doxorubicin against HCC827-ErlR cells was 0.13 μM, which is 21 times higher than that of HCC827 parental cells (Table 1), indicating that HCC827-ErlR cells were resistant to multiple anticancer drugs (erlotinib, gefitinib, osimertinib, and doxorubicin). In particular, doxorubicin is a cytotoxic chemotherapy drug, and resistance to doxorubicin indicates that HCC827-ErlR cells may have multiple EGFR-independent resistance mechanisms.

To analyze the temporal change in protein expression in response to erlotinib, HCC827 and HCC827-ErlR cells were treated with erlotinib at different concentrations for 48 h. As shown in Figure 1C, when HCC827 cells were treated with 45 nM erlotinib, phosphorylation of EGFR, MEK1/2, and ERK1/2 was decreased by more than 80% compared with that in vehicle-treated cells. When HCC827-ErlR cells were treated with 45 nM erlotinib for 48 h, EGFR phosphorylation was reduced by 90%. This result suggested that erlotinib resistance was independent of drug efflux, since the phosphorylation of EGFR was effectively suppressed by erlotinib in HCC827-ErlR cells (Figure 1C). However, the phosphorylation of MEK1/2 and ERK1/2 was decreased by 40% and 20%, respectively, compared with that in vehicle-treated cells. These results suggested that HCC827-ErlR cells may express different kinase(s) responsible for MEK1/2 phosphorylation and that AXL kinase may be an alternative kinase for the phosphorylation of MEK1/2. The phosphorylation of HSF1 at Ser326 was significantly inhibited by 45 nM erlotinib in HCC827 cells but not in HCC827-ErlR cells. HSF1 was significantly phosphorylated even with 1 μM erlotinib in HCC827-ErlR cells.

Similarly, when PC9 cells were treated with 45 nM erlotinib for 48 h, the phosphorylation of EGFR, MEK1/2, ERK1/2, and HSF1 was decreased by more than 70% compared with that in vehicle-treated cells (Figure 1D). However, when PC9-ErlR cells were treated with 45 nM erlotinib for 48 h, the phosphorylation of EGFR (T790M) was not decreased. Similarly, the phosphorylation of MEK1/2, ERK1/2, and HSF1 was slightly reduced compared with that in vehicle-treated cells. Treatment of PC9-ErlR cells with 3 μM erlotinib reduced the phosphorylation of EGFR by 30%, indicating that erlotinib may weakly inhibit EGFR (T790M) in PC9-ErlR cells. As EGFR (T790M)-targeting osimertinib is currently used in the clinic, we decided to focus our efforts to identify potential targets for overcoming multiple resistance mechanisms in HCC827-ErlR cells.

The epithelial-mesenchymal transition (EMT) has been reported to be associated with drug resistance (for review, [43]). Although HCC827 cells expressed the epithelial cell marker E-cadherin, HCC827-ErlR cells expressed mesenchymal cell markers such as vimentin and N-cadherin (Figure 1E), indicating that HCC827-ErlR cells underwent the EMT. In addition, HCC827-ErlR cells highly expressed AXL receptor tyrosine kinase compared with HCC827 cells. As expected, HCC827-ErlR cells exhibited a higher migration and invasion capacity than HCC827 cells (Figure 1F,G).

### 3.2. CMap Analysis of the Transcriptional Signature for Resistance to EGFR TKIs

We performed transcriptome analysis on HCC827 and HCC827-ErlR cells and obtained a list of 199 upregulated and 208 downregulated genes. The RNA-seq dataset was deposited in the NCBI’s Sequence Read Archive (accession number: PRJNA730670). We used a false discovery rate (FDR) < 0.001 and fold change of ≥4 or ≤−6.5. HCC827-ErlR cells significantly increased the mRNA expression of mesenchymal markers, such as VIM (24-fold), AXL (66-fold), and ZEB1 (9-fold) (Appendix A).

The CMap developed by the Broad Institute was used to identify potential target proteins and drugs that could reverse erlotinib resistance in HCC827-ErlR cells. We submitted a query of 407 differentially expressed genes (DEGs) to the CMap database for analysis (Figure 2).

From this analysis, the top four compounds (monorden, geldanamycin, cephaeline, and CP-863187) were selected based on the connectivity rank (Table 2).

The first (Monorden) and second (geldanamycin) compounds are HSP90 inhibitors and have positive enrichment scores. The third (cephaeline) and fourth (CP-863187) have negative enrichment scores.

Monorden (radicicol) and geldanamycin have positive mean values of 0.345 and 0.399, respectively, indicating that these drugs can induce a gene expression signature similar to HCC827-ErlR cells. In contrast, cephaeline and CP-863187 had negative mean values of −0.488 and −0.331, respectively, suggesting that these compounds could reverse the resistance phenotype. Monorden and geldanamycin activate HSF1 by inhibiting HSP90, releasing HSF1 from the HSP90/HSF1 complex [44,45]. In contrast, cephaeline and CP-863187 inhibit HSF1 activity [46,47]. Taken together, these results suggested that HSF1 is a molecule that targets erlotinib resistance and that cephalin and CP-863187 might be HSF1 inhibitors to overcome this resistance.

### 3.3. HSF1 Is a Potential Target to Overcome EGFR-TKI Resistance

Since HSF1 is a potential target to overcome drug resistance, we decided to analyze the proliferation and clonogenic activity of HCC827-ErlR cells after the inhibition of HSF1 by using two specific HSF1 shRNAs. As shown in Figure 3A,B, the knockdown of HSF1 in HCC827-ErlR cells effectively decreased proliferation 3 days after transduction and reduced colony formation 5 days after transfection. When HSF1 was downregulated by shRNAs in HCC827-ErlR cells, the amount of HSF1 and its downstream proteins, such as HSP70 and HSP27, was decreased (Figure 3C). In addition, HSP90/HSP70/BAG3 client proteins, such as EGFR, BCL2, and MCL1, were decreased, and PARP cleavage was induced, indicating apoptosis. Activation of caspase-3 and cleavage of PARP was detected 3 days after transfection, and further incubation increased the amount of active caspase-3 (Figure 3C). However, the amount of cleaved PARP was decreased, indicating further degradation of the cleaved PARP. Furthermore, the amount of the mesenchymal marker vimentin was decreased by HSF1 shRNAs, suggesting that HSPs may be involved in the EMT phenotype. Notably, HSF1 knockdown using specific shRNAs efficiently decreased the amount of EGFR protein.

Next, we decided to use a specific HSF1 inhibitor to confirm the results obtained with HSF1 shRNAs. Previously, we reported that KRIBB11 binds and inhibits HSF1 transcription factor activity [42]. We wanted to determine the effect of KRIBB11 on the proliferation of HCC827 and HCC827-ErlR cells. Thus, HCC827 and HCC827-ErlR cells were treated with KRIBB11 at different concentrations (0–100 μM) for 72 h. KRIBB11 exhibited a dose-dependent inhibition of HCC827 and HCC827-ErlR cell growth at a broad range of concentrations, and the GI_50_ values of KRIBB11 were 2.8 μM and 1.4 μM, respectively, indicating that HCC827-ErlR cells were slightly more sensitive to KRIBB11 than HCC827 cells (Figure 4A).

In addition, cells were stained with annexin-V-FITC and propidium iodide to analyze the effect of KRIBB11 on apoptosis. As shown in Figure 4B, the percentage of apoptotic cells increased to approximately 11% and 14% by incubating HCC827 and HCC827-ErlR cells with 5 μM KRIBB11, which is consistent with PARP cleavage.

To analyze the temporal change in protein expression by KRIBB11, HCC827-ErlR cells were treated with KRIBB11 at 10 and 20 μM for different time periods. As shown in Figure 4C, KRIBB11 effectively downregulated the amount of HSP70 and BAG3 in a time- and dose-dependent manner. Since the HSP70/BAG3 complex is a molecular chaperone that stabilizes antiapoptotic BCL2 family proteins, we assessed the effects of KRIBB11 on BCL2 and MCL1 protein levels. As expected, the amounts of BCL2 and MCL1 were reduced by KRIBB11. Activation of caspase-3 and cleavage of PARP were detected 24 h after the treatment and further incubation increased the amount of active caspase-3 (Figure 4C). However, the amount of cleaved PARP was decreased, indicating the further degradation of cleaved PARP. Of note, MCL1 significantly decreased 12 h after treatment of cells with KRIBB11 compared to BAG3. KRIBB11 has been reported to inhibit MCL1 in an HSF1/BAG3-dependent and HSF1/BAG3-independent manner, which may explain this [48].

Multiple mechanisms of erlotinib resistance have been reported. These include compensatory activation of receptor tyrosine kinases (RTKs), such as MET and AXL, or a secondary gatekeeper EGFR (T790M) mutation. Interestingly, KRIBB11 decreased the amount of EGFR, MET, and AXL in HCC827-ErlR cells in a time- and dose-dependent manner (Figure 4C), suggesting that KRIBB11 induced apoptosis of HCC827-ErlR cells by simultaneously inhibiting HSP70/BAG3/BCL2/MCL1 survival signaling and decreasing the levels of EGFR/MET/AXL receptor tyrosine kinase proteins. Similarly, when we analyzed the temporal change in protein expression in HCC827 parental cells, KRIBB11 decreased the HSP70/HSP27/BCL2/MCL1 and EGFR/MET levels, inducing caspase-3 activation and PARP cleavage (Figure 4D). Cleavage of PARP appeared 12 h after the treatment, peaked at 24 h after treatment, and gradually decreased with further incubation, but the amount of active caspase-3 was positively correlated with the duration of treatment.

### 3.4. KRIBB11 Downregulates EGFR (T790M), AXL, MET, HSP70, HSP27, BAG3, and MCL1 in NCI-H820 and PC9-ErlR Cells

We wanted to test whether KRIBB11 could induce apoptosis in cancer cells that had acquired erlotinib and osimertinib resistance in cancer patients. NCI-H820 cells were isolated from Caucasian male patients and reported to have EGFR (T790M), MET, and AXL [19,49]. As shown in Figure 5A, the GI_50_ values of erlotinib and osimertinib in the NCI-H820 cells were 1.16 μM and 8.17 μM, respectively, indicating that the NCI-H820 cells are first-generation erlotinib- and third-generation osimertinib-resistant cancer cells.

To analyze the temporal changes in protein expression induced by KRIBB11, NCI-H820 cells were treated with 10 µM and 20 µM KRIBB11 for different time periods. As shown in Figure 5B, when NCI-H820 cells were treated with KRIBB11 for 12 h, the expression of HSP27 decreased, and the downregulation of HSP70, MCL1, EGFR, and MET was observed 24 h after the treatment. Significant downregulation of HSP70/BAG3/MCL1 and EGFR/MET was consistent with PARP cleavage, indicating apoptosis of NCI-H820 cells. Collectively, these results suggested that HSF1 inhibition could be useful in treating erlotinib- and osimertinib-resistant NSCLC patients.

Since NCI-H820 cells expressing EGFR (T790M) and MET were effectively suppressed by KRIBB11, we also tested whether PC9-ErlR cells expressing EGFR (T790M) were also sensitive to KRIBB11. As shown in Figure 5C, the GI_50_ values of erlotinib and KRIBB11 in PC9-ErlR cells were 3.98 μM and 3.15 μM, respectively. When PC9-ErlR cells were treated with KRIBB11 at 10 and 20 µM for 12 h, the amount of HSP27, BAG3, and MCL1 was decreased, and HSP70 and EGFR (T790M) were downregulated 24 h after the treatment (Figure 5D). The downregulation of HSP70/BAG3/MCL1/EGFR (T790M) is consistent with PARP cleavage 48 h after treatment. Taken together, KRIBB11 decreased HSF1 downstream proteins, such as HSP70 and HSP27, and HSP90/HSP70 client proteins, such as EGFR, MET, and AXL, causing apoptotic cell death.

### 3.5. Emetine Decreased the Growth of PC9-ErlR-Resistant Cells in BALB/C Nude Mice

KRIBB11 effectively decreased the HSP70/BAG3/BCL2 and EGFR/MET/AXL proteins, leading to apoptosis of resistant cancer cells. However, KRIBB11 is a reagent for studying HSF1 functions and cannot be used in the clinic. Therefore, for use in the clinic in the short term, we used a drug repositioning approach to find drugs that can inhibit HSF1 activity. Interestingly, cephaeline was reported to suppress HSF1 activity by inhibiting translation elongation [46]. Emetine has the same structure as cephaeline, except for one side chain [50,51], and is an antiprotozoal drug (Figure 6A).

Therefore, we decided to analyze the HSF1 inhibitory activity of cephaeline and emetine. To accomplish this, HCT116 cancer cells were cotransfected with a p(HSE)4-TA-Luc reporter and an internal control vector constitutively expressing pRL-TK to normalize for transfection efficiency. Then, the cells were treated with cephaline or emetine for 30 min and exposed to heat shock at 44 °C for 15 min. After 5 h of recovery at 37 °C, luciferase activity was measured. From this, we found that cephaline and emetine inhibited HSF1 reporter activity (Figure 6B). Consistent with its effect on the HSF1 reporter, cephaline and emetine also significantly downregulated HSF1 downstream proteins, including HSP70, HSP27, and BAG3, in a concentration-dependent manner (Figure 6C). In addition, emetine blocked heat-induced HSP90α expression. As emetine showed potent HSF1 inhibitory activity and is relatively inexpensive compared with cephaline, we decided to use emetine in vivo as an HSF1 inhibitor.

To confirm the efficacy of emetine on multiple mechanisms of drug resistance, we performed tumor xenograft studies in mice. Unfortunately, HCC827-ErlR cells failed to grow as xenografts and therefore could not be used for in vivo study. Similar tumor growth failure with HCC827 mesenchymal cells was previously reported by another group [52]. Next, we tried to use patient-derived EGFR TKI-resistant NCI-H820 cancer cells for the tumor xenograft assay, but we were unable to detect tumor growth even 3 months after mouse inoculation. Indeed, we could not find any research papers for tumor xenograft studies using NCI-H820 cancer cells. In the end, we were able to conduct tumor xenograft studies using erlotinib-resistant PC9-ErlR cells expressing the EGFR (T790M) mutation. PC9-ErlR cells were treated with 0.5 μM and 1 μM emetine for different time periods. As shown in Figure 6D, treating PC9-ErlR cells with emetine for 24 h decreased the expression of HSP70, BAG3, HSP27, BCL2, MCL1, and EGFR, and PARP cleavage was observed 48 h after the treatment, indicating apoptotic cell death.

PC9-ErlR tumor xenografts in BALB/C nude mice were used to investigate the antitumor effects of emetine. Flanks of each mouse were injected with 9 × 10^6^ cells. When the mean tumor volume reached 55 mm^3^, we administered 1 mg/kg of emetine for 5 days, and beginning on day 7, we administered 10 mg/kg of emetine 5 days per week for 18 days (Figure 6E). Mice treated with emetine showed a 77.4% (*p* < 0.05) decrease in tumor volume and a 73.2% (*p* < 0.05) decrease in tumor weight compared with control mice (Figure 6F). These results suggest that emetine could block the growth of erlotinib-resistant tumors in an animal model. When emetine was used at 10 mg/kg, body weight changed by less than 10% (Figure 6E).

To determine whether emetine suppressed the growth of PC9-ErlR tumors through the inhibition of HSF1 in vivo, we measured the protein levels of HSF1, EGFR, HSP27, and MCL1 in tumor tissues of emetine- and control-treated mice. As shown in Figure 6G,H, EGFR, HSF1, HSP27, and MCL1 were significantly decreased in tumors from mice treated with emetine compared with mice treated with the vehicle. Although the results were not statistically significant, HSF1 activity was inhibited by emetine in vivo.

## 4. Discussion

Based on CMap analysis, we speculated that HSF1 could be a potential target to overcome erlotinib resistance. To prove the therapeutic effects of HSF1 inhibition, we decided to use two specific HSF1 shRNAs. As shown in Figure 3C, the knockdown of HSF1 expression by specific shRNAs decreased the amount of HSF1 protein and its downstream proteins, such as HSP70 and HSP27. In addition, the reduction of HSP70/BAG3 chaperone complexes decreased its client proteins, such as BCL2 and MCL1. Surprisingly, we also observed the downregulation of HSP90/HSP70 client proteins, such as EGFR (Figure 3C). We also confirmed these results using the HSF1 inhibitor compound KRIBB11 (Figure 4C). Collectively, the treatment of HCC827 and HCC827-ErlR cells with an HSF1 inhibitor decreased not only the amounts of HSF1 downstream proteins, such as HSP70, HSP27, and BAG3, but also decreased HSP90/HSP70/BAG3 client proteins, such as BCL2, MCL1, EGFR, MET, and AXL.

Patient tumors simultaneously acquire multiple mechanisms of resistance [18,53,54,55]. In most NSCLC patient cases, the first-line treatment of EGFR-TKIs lose their therapeutic activity within 13 months due to acquired resistance, which is mediated by EGFR target alterations, such as the secondary EGFR (T790M) mutation, by activation of bypass signaling, such as HER2 amplification, MET amplification, PIK3CA, MAPK, BRAF, and multiple signaling pathways, or by phenotypic changes, such as SCLC transformation and the EMT (for review, [56,57]). Progression-free survival on the osimertinib first line is 18.9 months versus 10.2 months for other first-generation EGFR-TKIs [15]. A complete response is achieved only in a small number of patients (less than 5%) after EGFR-TKI treatment [14,15]. Therefore, targeting multiple resistance signaling pathways is critical to overcome EGFR-TKI resistance, and inhibition of HSF1 can be a great approach to block multiple signaling pathways simultaneously (Figure 6I).

Lee and colleagues reported that the inhibition of EGFR kinase activity by erlotinib could not inhibit the proliferation of H1650, NCI-H1975, or NCI-H820 cells, whereas the depletion of EGFR protein by specific shRNAs almost completely blocked the growth of these cells, indicating that EGFR protein maintained cell viability in a kinase activity-independent manner [58]. This result suggested that depleting the EGFR protein itself should be a better strategy to overcome EGFR-TKI resistance. Interestingly, when we treated HCC827-ErlR cells with HSF1 inhibitors, such as HSF1 shRNA or KRIBB11, the amount of EGFR protein was significantly reduced (Figure 3C and Figure 4C).

We wanted to confirm the efficacy of HSF1 inhibitors on cancer cells that have multiple resistance mechanisms. To accomplish this, we used the NCI-H820 lung cancer cell line, which has acquired multiple drug resistance mechanisms. NCI-H820 cancer cells express an EGFR drug-sensitive mutation (an exon 19 deletion), an EGFR drug-resistance mutation (T790M), activated AXL, and MET amplification, conferring resistance to erlotinib, gefitinib, and afatinib [19,49]. In addition, we found that NCI-H820 cells acquired an osimertinib-resistant phenotype. When we treated NCI-H820 cells with KRIBB11, the levels of HSF1 downstream proteins, such as HSP70, HSP27, and BAG3, and HSP90/HSP70/BAG3 client proteins, such as BCL2, MCL1, EGFR, and MET, were reduced, inducing PARP cleavage (Figure 5B). These results confirmed the effectiveness of inhibiting HSF1 to overcome both first-generation TKI erlotinib resistance and third-generation TKI osimertinib resistance.

HCC827-ErlR cells had a mesenchymal morphology, expressed mesenchymal markers, such as vimentin, and obtained higher migration and invasion activity. The EMT has been reported to be associated with drug resistance (for review, [43]). The EMT signature of 76 genes predicted resistance to EGFR inhibitors, and AXL was identified as a therapeutic target for overcoming EGFR-TKI resistance [59]. Interestingly, KRIBB11 reduced the amount of AXL protein (Figure 4C), confirming that KRIBB11 can be used to overcome EMT-mediated TKI resistance. In addition, HSF1 inhibition by HSF1 shRNAs or KRIBB11 decreased EMT marker proteins, such as vimentin and N-cadherin (Figure 3C and Figure 4C).

The connectivity map (CMap) is a big data collection of the transcriptome and has become an economic tool for the identification of biologically active compounds without biological screening experiments [60,61]. As cephaeline and CP-863187 have negative enrichment scores, we speculated that these compounds have the potential to reverse erlotinib resistance. Santagate and colleagues reported that protein translation and HSF1 activation were coordinated to support anabolic metabolism in malignant cancer cells [46] and that cephaeline is a translation inhibitor. Emetine has the same structure as cephaeline, except for one side chain [50,51]. We showed that cephaeline and emetine inhibited HSF1 reporter activity (Figure 6B) and the expression of HSF1 downstream proteins, such as HSP90α, HSP70, HSP27, and BAG3 proteins (Figure 6C). In particular, emetine is a drug used in the clinic as an antiprotozoal and is relatively inexpensive compared with cephaline. Therefore, we chose to use emetine for in vivo experiments. We expected that emetine may reduce the tumor growth of PC9-ErlR cells because these cells acquired erlotinib resistance through the EGFR (T790M) mutation. We also expected that emetine could reduce the EGFR (T790M) mutant protein level (Figure 6D). As shown in Figure 6E,G,H, the administration of emetine decreased tumor growth by 77% with reduced amounts of EGFR and MCL1 proteins.

We tested whether there was any synergistic effect of the combination of emetine with erlotinib in erlotinib-resistant HCC827-ErlR cells. The proliferation of HCC827-ErlR cells was inhibited to a similar extent when emetine was administered alone or in combination with erlotinib. As treating the HCC827-ErlR cells with emetine decreased the amount of EGFR protein, the additional inhibition of EGFR with erlotinib probably could not exert its inhibitory effect.

HCC827-ErlR cells showed resistance to doxorubicin compared with HCC827 cells (Table 1). Doxorubicin induces the expression of multidrug resistance-associated protein 1 (MDR1), which mediates doxorubicin efflux [62]. Interestingly, HSF1 modulates MDR1/P-gp expression at the transcript level, resulting in a multidrug resistance phenotype [63,64]. Thus, HSF1 inhibition could decrease the possibility of MDR1-mediated drug resistance.

RTKs responsible for EGFR-TKI resistance, including EGFR, MET, and AXL, are known HSP90 clients. There were several reports that HSP90 inhibitors could decrease RTKs, overcoming drug resistance [65,66,67]. Although HSP90 inhibitors decrease HSP90 client proteins, they activate the HSF1-dependent stress response and thus paradoxically induce HSP90, HSP70, HSP27, BAG3, and MDR1, conferring drug resistance. Therefore, HSP90 inhibitors have limited efficacy, and HSF1 inhibition may be a better strategy to block RTK-mediated drug resistance.

## 5. Conclusions

In this study, we identified HSF1 as a potential target protein for overcoming the resistance to EGFR-TKI. We proved that HSF1 inhibition by using HSF1 shRNAs, KRIBB11, or emetine decreased the expression of HSF1 downstream proteins, such as HSP70 and HSP27, and also decreased the amount of HSP90/HSP70/BAG3 client proteins, such as BCL2, MCL1, EGFR, MET, and AXL, blocking multiple drug resistance mechanisms. In addition, we proved the efficacy of HSF1 inhibitors to induce apoptosis of several EGFR-TKI-resistant cell lines in vitro and to inhibit the tumor growth of PC9-ErlR cells in vivo. Therefore, we concluded that HSF1 is a promising therapeutic target and KRIBB11 or emetine can be used as a lead compound to develop a therapeutic drug for overcoming EGFR-TKIs resistance in NSCLC.

## Figures and Tables

**Figure 1 cancers-13-02987-f001:**
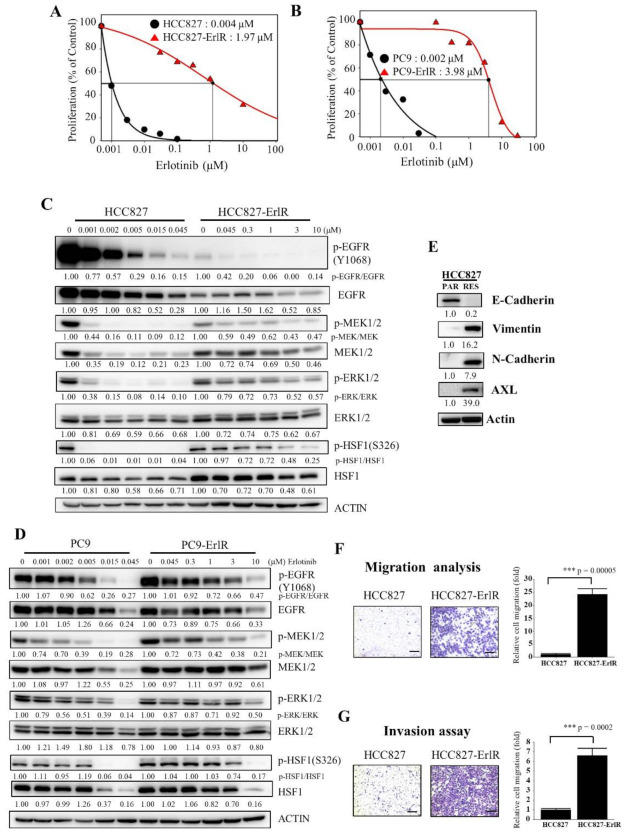
Generation and characterization of erlotinib-resistant NSCLC cancer cell lines. (**A**,**B**) HCC827, HCC827-ErlR, PC9, and PC9-ErlR cells were treated with 0.1% DMSO or different concentrations of erlotinib. After incubation for 72 h, cells were counted using a microscope. Proliferation is expressed as the percentage of erlotinib-treated cells compared with 0.1% DMSO-treated cells. Each value is the mean ± S.D. (**C**) HCC827 and HCC827-ErlR cells were treated with 0.1% DMSO or various concentrations of erlotinib for 48 h, and whole cell lysates were analyzed by Western blotting (the protein band intensity was normalized to the actin band intensity, and the phosphorylated protein band intensity was normalized to the total protein intensity). (**D**) PC9 and PC9-ErlR cells were treated with 0.1% DMSO or various concentrations of erlotinib, and whole cell lysates were analyzed by Western blotting (the protein band intensity was normalized to the actin band intensity). (**E**) Western blot analysis of HCC827 and HCC827-ErlR cells was conducted to assess epithelial and mesenchymal marker proteins using the indicated antibodies (the protein band intensity was normalized to the actin band intensity). (**F**) Migration assays of HCC827 and HCC827-ErlR cells were performed for 18 h using Transwells. Cells attached to the lower surface of the membrane were stained with crystal violet and visualized under a light microscope. Migrated cells were quantified with Image-ProPlus 5.0 software. Scale bars, 200 μM. The data represent the mean ± S.D.; comparisons were performed with *t*-tests (two groups); *** *p* < 0.001. (**G**) Invasion assays of HCC827 and HCC827-ErlR cells were performed for 18 h. Scale bars, 200 μM. The data represent the mean ± S.D.; comparisons were performed with *t*-tests (two groups); *** *p* < 0.001.

**Figure 2 cancers-13-02987-f002:**
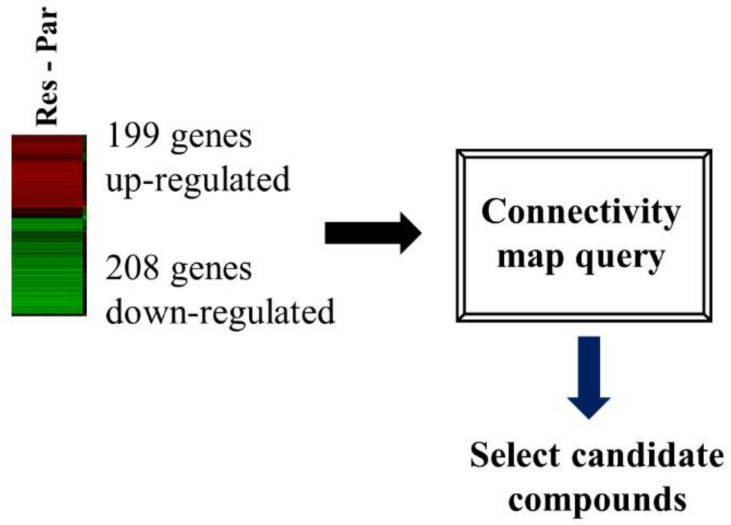
HSF1 was chosen as a potential target protein to overcome erlotinib resistance. A flow chart is shown of the CMap analysis procedure that was used to determine whether cephaeline could reverse erlotinib resistance. Gene expression profiling of RNA-seq was used to select DEGs for the erlotinib resistance phenotype. In HCC827-ErlR cells, 199 4-fold upregulated genes and 208 6.5-fold downregulated genes were selected for CMap analysis.

**Figure 3 cancers-13-02987-f003:**
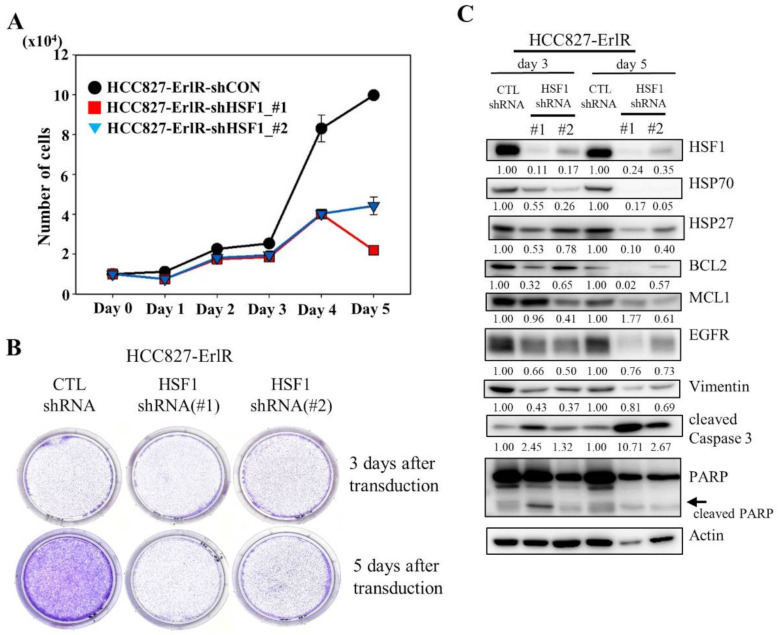
The shRNA-mediated knockdown of HSF1 inhibits the proliferation and colony formation of HCC827-ErlR cells. (**A**) HCC827-ErlR cells were transduced with lentivirus carrying shRNA targeting HSF1 (#1 or #2) or control shRNA. Twenty-four hours after transduction (day 0), cells were counted using a microscope on the designated days. (**B**) Representative images are shown of colonies formed by HCC827-ErlR cells transduced with control shRNA, HSF1 shRNA (#1), or HSF1 shRNA (#2) after 3 or 5 days of transduction. (**C**) HCC827-ErlR cells were transduced with control shRNA, HSF1 shRNA (#1), or HSF1 shRNA (#2), and whole cell lysates were prepared 3 and 5 days after transduction. Expression of HSF1 or its downstream proteins was analyzed by Western blotting using the indicated antibodies (protein band intensity was normalized to the actin band intensity).

**Figure 4 cancers-13-02987-f004:**
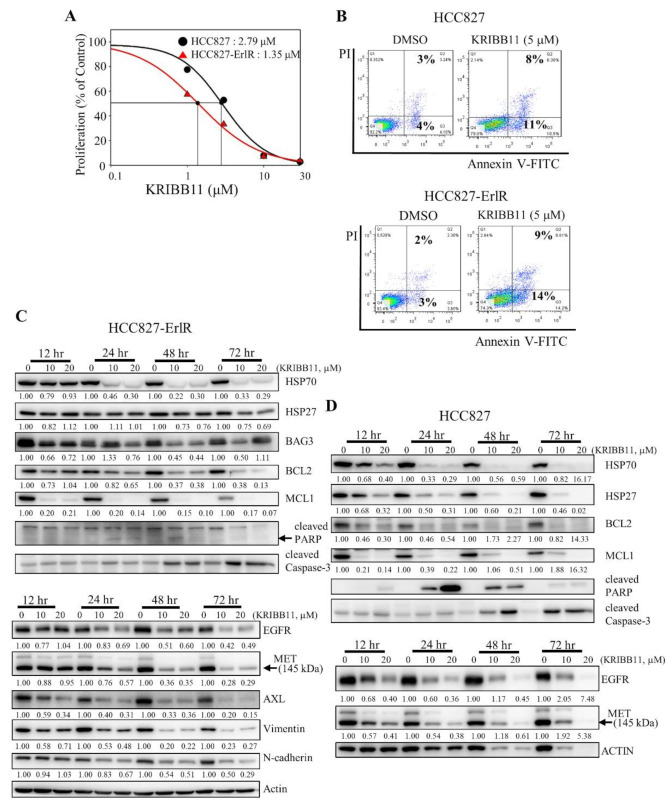
The HSF1 inhibitor KRIBB11 inhibits the proliferation of HCC827 and HCC827-ErlR cells and induces apoptosis. (**A**) HCC827 and HCC827-ErlR cells were treated with 0.1% DMSO or different concentrations of KRIBB11. After incubation for 72 h, cells were counted using a microscope. Proliferation is expressed as the percentage of KRIBB11-treated cells compared with 0.1% DMSO-treated cells. Each value is the mean ± S.D. (**B**) HCC827 and HCC827-ErlR cells treated for 48 h with 0.1% DMSO or 5 μM KRIBB11 were analyzed by annexin-V/FITC/PI flow cytometry. (**C**,**D**) HCC827-ErlR (**C**) and HCC827 (**D**) cells were treated with 0.1% DMSO or KRIBB11 (10, 20 μM) for 12–72 h, and whole cell lysates were analyzed by Western blotting with antibodies as indicated (the protein band intensity was normalized to the actin band intensity).

**Figure 5 cancers-13-02987-f005:**
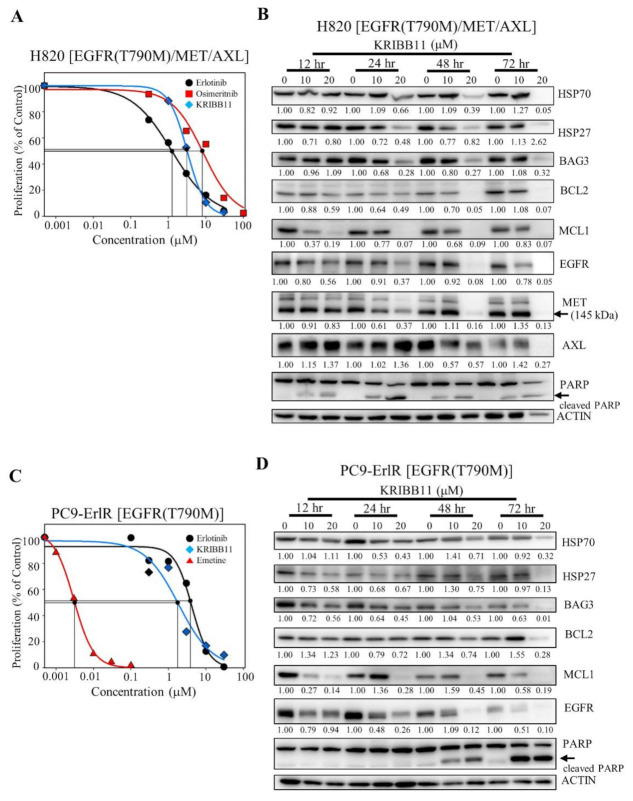
KRIBB11 decreases the amount of HSF1 downstream proteins, such as HSP70, and HSP70/HSP90 client proteins such as EGFR. (**A**) NCI-H820 cells were treated with 0.1% DMSO or different concentrations of erlotinib, osimertinib, and KRIBB11. After incubation for 72 h, cells were counted using a microscope. Proliferation is expressed as the percentage of erlotinib-, osimertinib-, and KRIBB11-treated cells compared with 0.1% DMSO-treated cells. Each value is the mean ± S.D. (**B**) NCI-H820 cells were treated with 0.1% DMSO or KRIBB11 (10, 20 μM) for 12–72 h, and whole cell lysates were analyzed by Western blotting using the indicated antibodies (the protein band intensity was normalized to the actin band intensity). (**C**) PC9-ErlR cells were treated with 0.1% DMSO or different concentrations of erlotinib, KRIBB11, and emetine. After incubation for 72 h, cells were counted using a microscope. Proliferation is expressed as the percentage of erlotinib-, KRIBB11-, and emetine-treated cells compared with 0.1% DMSO-treated cells. Each value is the mean ± S.D. (**D**) PC9-ErlR cells were treated with 0.1% DMSO or KRIBB11 (10, 20 μM) for 12–72 h, and whole cell lysates were analyzed by Western blotting using the indicated antibodies (the protein band intensity was normalized to the actin band intensity).

**Figure 6 cancers-13-02987-f006:**
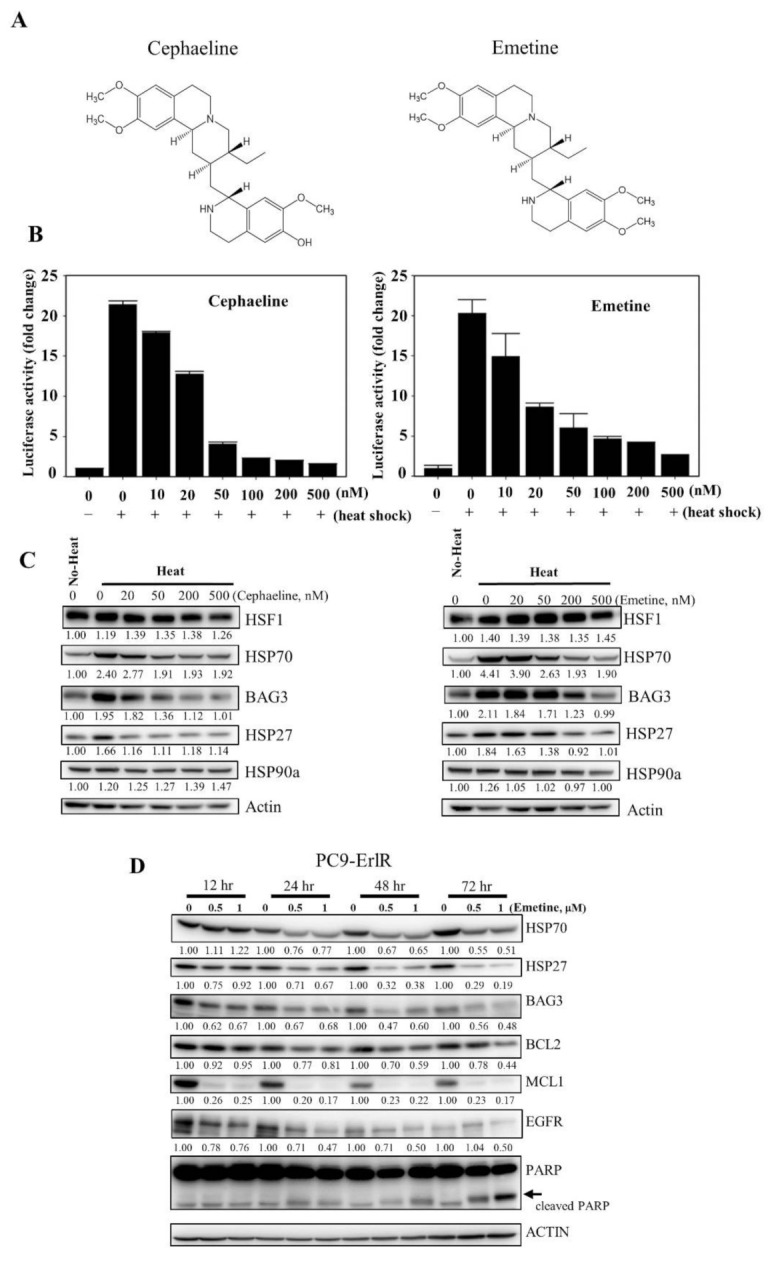
Emetine inhibits HSF1 activity and growth of PC9-ErlR cells in xenograft models. (**A**) The cephaeline and emetine structures are shown. (**B**) HCT116 cells were transfected with the p(HSE)4-TA-Luc reporter plasmid and treated with or without heat in the presence of different concentrations (10–500 nM) of cephaeline and emetine. Reporter assay was performed as described in Materials and Methods. (**C**) Cephaeline and emetine inhibited heat-induced expression of HSP70, BAG3, and HSP27 in HCT-116 cells. HCT-116 cells were treated with the indicated concentrations of cephaeline and emetine for 30 min, exposed to heat shock at 42 °C for 30 min, and then incubated for an additional 5 h at 37 °C. Whole-cell lysates were analyzed by Western blotting using the indicated antibodies (the protein band intensity was normalized to the actin band intensity). (**D**) PC9-ErlR cells were treated with 0.1% DMSO or the indicated concentration of emetine for 12–72 h. Whole-cell lysates were subjected to Western blotting (the protein band intensity was normalized to the actin band intensity); (**E**) To evaluate the in vivo antitumor activity of emetine, PC9-ErlR cells were implanted subcutaneously into the right flank of nude mice. When the tumor volume reached 55 mm^3^ (day 1), PBS or emetine (1 mg/kg) was injected intraperitoneally for 5 days. Beginning on day 7, emetine (10 mg/kg) was injected intraperitoneally for 5 days per week for 18 days. Mice were sacrificed on day 25. The results shown are from one assay using 12 mice (6 mice for each compound). Tumor volumes were estimated using the formula: length (mm) × width (mm) × height (mm)/2. Body weight was measured on each indicated day. (**F**) The final tumor weights were measured. (**G**) For each compound, lysates were prepared from tumor tissues of 6 mice, and whole cell lysates were analyzed by Western blotting using the indicated antibodies (the protein band intensity was normalized to the actin band intensity). (**H**) The relative abundance of EGFR, HSF1, HSP27, and MCL1 in tumors was quantified using the LAS-4000 mini. (**I**) The proposed mechanism is shown by which HSF1 inhibition overcomes multiple mechanisms of EGFR-TKI resistance.

**Table 1 cancers-13-02987-t001:** The sensitivity of erlotinib-resistant and parental HCC827 and PC9 cells treated with anticancer drugs. GI_50_ values of HCC827, HCC827-ErlR, PC9, and PC9-ErlR cells treated with erlotinib, gefitinib, osimertinib, and doxorubicin are shown. GI_50_ values were calculated using SigmaPlot software.

	GI50 (μM)			GI50 (μM)		
	HCC827	HCC827-ErlR	Ratio	PC9	PC9-ErlR	Ratio
Erlotinib	0.004	1.97	492.5	0.002	3.98	1990
Gefitinib	0.004	21.74	5435	0.01	8.37	837
Osimertinib	0.016	3.22	201.25	0.0006	0.003	5
Doxorubicin	0.006	0.13	21.67	0.037	0.059	1.59

**Table 2 cancers-13-02987-t002:** The top 4 compounds were selected based on connectivity ranking.

Rank	CMap Name	Mean	n	Enrichment	*p*	Specificity	Percent Non-Null	Function
1	Monorden	0.345	22	0.524	0	0.0267	54	HSP90 inhibitor
2	Geldanamycin	0.399	15	0.58	0.00004	0.1033	60	HSP90 inhibitor
3	Cephaeline	−0.448	5	−0.761	0.00146	0.1265	80	Translation inhibitor
4	CP-863187	−0.331	4	−0.829	0.00159	0	75	p38 MAPK inhibitor

## Data Availability

The RNA-seq dataset obtained in this study was deposited in the NCBI’s Sequence Read Archive (accession number: PRJNA730670) and is available.

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
