# Peer review of "Targeting HSF1 as a Therapeutic Strategy for Multiple Mechanisms of EGFR Inhibitor Resistance in EGFR Mutant Non-Small-Cell Lung Cancer"

_cancers, 2021, doi:10.3390/cancers13122987_

Round 1
Reviewer 1 Report
Dear doctor Lee and colleagues,
Thank you for very interesting article describing the first time in details the mechanism of HSF1 action. I think that your work contributes a lot in our understanding of the role of HPF1 in EGFR-TKI resistant cells.
Please find attached my questions and remarks, thank you:
Questions:
1. If we assume that HSF1 is overproduced in progressing tumor cells, how do you suggest, which assay we can use to reliably measure HSF1 status in rebiopsy samples of EGFR-TKI resistant NSCLC patients?
2. Based on your current work, do you think that do we already have a solid basis for adopting HSF1 as a prognostic and diagnostic biomarker for EGFR-mutated NSCLC? It was established in other malignancies (e.g., Kim, W., Kim, S.-J., 2021. Heat Shock Factor 1 as a Prognostic and Diagnostic Biomarker of Gastric Cancer. Biomedicines 9, 586. doi:10.3390/biomedicines9060586, and Wan T, Shao J, Hu B, Liu G, Luo P, Zhou Y. Prognostic role of HSF1 overexpression in solid tumors: a pooled analysis of 3,159 patients. Onco Targets Ther. 2018 Jan 17;11:383-393. doi: 10.2147/OTT.S153682).
- Why have you chosen Doxorubicin for HCC827-ErlR cells instead of platin, which is otherwise routinely used for NSCLC?
- What was the reason that you have not used CP-863187 in your experiments?
- Do you expect any challenges while using KRIBB11 or Emetine in future clinical trials?
Remarks:
Line 24, 25 – first time KRIBB11 used, please expand the acronym: N2-(1H-indazol-5-yl)-N6-methyl-3-nitropyridine-2,6-diamine, and mention about is role, e.g., KRIBB11 abolishes the heat shock-induced luciferase activity. Please also expand the acronym for HSF1.
Line 56 and line 570 – It is not correct. Progression free survival on Osimertinib first line is 18.9 months versus 10.2 month for first generation EGFR-TKI. ( Soria JC, Ohe Y, Vansteenkiste J, Reungwetwattana T, Chewaskulyong B, Lee KH, Dechaphunkul A, Imamura F, Nogami N, Kurata T, Okamoto I, Zhou C, Cho BC, Cheng Y, Cho EK, Voon PJ, Planchard D, Su WC, Gray JE, Lee SM, Hodge R, Marotti M, Rukazenkov Y, Ramalingam SS; FLAURA Investigators. Osimertinib in Untreated EGFR-Mutated Advanced Non-Small-Cell Lung Cancer. N Engl J Med. 2018 Jan 11;378(2):113-125. doi: 10.1056/NEJMoa1713137). Otherwise, please specify, if you mean second or further line treatment with EGR-TKI.
Line 74 – acquired resistance includes also KRAS mutation (Urbanska et al. EGFR-L858R NSCLC with pleiotropic resistance mechanisms: T790M, C797S, SCLC-transformation and KRAS, TP53, and BRAF mutations. Journal of Thoracic Oncology 2021;16 (Suppl.)(3):S592-S593 abstr. P76.16).
Line 306 – In Tabl.1: either explain the abbreviation of Erl, Gef, Osim, Dox in the lines above, or provide the full names in the table.
Line 336 – expend the acronym EMT, as used the first time in the manuscript.
Line 362 – should it not be signed as Table 2?
Line 392 – you signed that figure as Fig. 1, but it should be Fig. 3, as you discuss in lines 376-390?
Line 410 – should it not be signed as a Fig.4 instead of 2?
Line 555 – please rephrase the sentence as it is unclear: “The trend suggested that HSF1 inhibition may be due to 4. Discussion”
Line 569 - you can start here the next chapter and call it Discussion for resuming the results mentioned above in the context of EGFR-TKI resistant mechanism.
Reviewer 2 Report
The authors attempted to identify target proteins and compounds that can be used to overcome EGFR-TKI resistance in NSCLC. To accomplish this, the authors generated EGFR inhibitor erlotinib-resistant HCC827-ErlR cells and identified HSF1 as a potential target protein to overcome erlotinib resistance. Using specific HSF1 shRNAs and KRIBB11, they proved the effectiveness of HSF1 inhibition for overcoming erlotinib resistance in vitro. In addition, the authors proved the efficacy of emetine in inhibiting HSF1 activity and the tumor growth of erlotinib-resistant PC9-ErlR cells in a mouse model.
However, the title of this manuscript, "Targeting HSF1 as a Therapeutic Strategy for Multiple Mechanisms of EGFR Inhibitor Resistance in EGFR Mutant Non-Small-Cell Lung Cancer" can not be justified by the result of this manuscript, since there are so many different mechanisms in the acquired EGFR Inhibitor resistance in EGFR Mutant Non-Small-Cell Lung Cancer.
As the authors stated multiple mechanisms of erlotinib resistance have been reported. These include compensatory activation of receptor tyrosine kinases (RTKs), such as MET and AXL, or a secondary gatekeeper EGFR (T790M) mutation. The 3rd generation EGFR-TKI have different resistance mechanisms. The effectiveness of HSF1 inhibition for overcoming erlotinib resistance was tested in erlotinib-resistant HCC827-ErlR cells, which has unknown mechanisms of erlotinib resistance and NCI H820 & PC9-ErlR cells, which has secondary gatekeeper EGFR (T790M) mutation. The efficacy of emetine in inhibiting HSF1 activity and the tumor growth was noted in erlotinib-resistant PC9-ErlR cells, which has secondary gatekeeper EGFR (T790M) mutation. The author's conclusion that they identified HSF1 as a potential target protein for overcoming the resistance to EGFR-TKI, can not be generalized to every resistance to EGFR-TKI nor different mechanisms in the acquired EGFR Inhibitor resistance.
Major comments:
- It would be better to add any clinical evidence of HSF1 expression and the prognosis of EGFR mutant NSCLC patients using public data.
- It would be better to test whether there is any synergistic effect of combination of emetine or cephaeline with erlotinib in erlotinib-resistant HCC827-ErlR cells.
Minor comments:
Erratum: erlotimib (line 26) -> erlotinib (line 26)
Erratum: Figure 1 (line 392) -> Figure 3 (line 392)
Erratum: Figure 2 (line 410) -> Figure 4 (line 410)
Round 2
Reviewer 2 Report
No specific comments.
This manuscript is a resubmission of an earlier submission. The following is a list of the peer review reports and author responses from that submission.